

# Two new *Cortinarius* species in subgenus *Leprocybe* from Southwest China

Peng Hong[1,2,*], Ke Wang[2,*], Zhuo Du[2], Ming-Jun Zhao[2], Meng-Le Xie[3], Di Liu[1] and Tie-Zheng Wei[2]

[1] Department of Horticulture and Landscape Architecture, College of Agriculture, Yanbian University, Yanji, Jilin, China
[2] State Key Laboratory of Mycology, Institute of Microbiology, Chinese Academy of Sciences, Beijing, China
[3] School of Food Science and Engineering, Yangzhou University, Yangzhou, Jiangsu, China
* These authors contributed equally to this work.

## ABSTRACT

Two new *Cortinarius* species in subgenus *Leprocybe*, *Cortinarius hengduanensis* and *C. yadingensi*s, are proposed based on a combination of morphological and molecular evidence. *Cortinarius hengduanensis* has distinct olive tinged basidiomata, a squamulose pileus, and small, subglobose to broadly ellipsoid basidiospores, the ITS sequence differs from that of *C. flavifolium* by at least 28 substitutions and independent positions. *Cortinarius yadingensis* has a squamulose pileus and subglobose to broadly ellipsoid coarsely verrucose basidiospores, the ITS sequence has at least 11 substitutions and index position deviations from the other members of the *Leprocybe* section. Both new species were found in mixed forests of southwest China.

## INTRODUCTION

*Cortinarius* (Pers.) Gray is the largest genus in Agaricales (Basidiomycota), usually forming ectomycorrhizas with members of the Pinaceae, Betulaceae, Fagaceae, Salicaceae and Ericaceae, and is distinguished from other agarics by arachnoid partial veil, together with rust brown and verrucose basidiospores (*Moser & Horak, 1975*). It is a complex genus with more than 5,900 names documented in Index Fungorum (http://www.indexfungorum.org/Names/Names.asp, data accessed on 12 April, 2024) and over 2,600 described species recorded in Catalogue of Life (https://www.catalogueoflife.org). The type locality of about 2,000 *Cortinarius* species were reported from Europe and North America (data derived from Fungal Names, https://nmdc.cn/fungalnames). However, its diversity and distribution in China are still poorly understood, as only 238 species reported in the Catalogue of Life China 2023 (*The Biodiversity Committee of Chinese Academy of Sciences, 2023*).

*Moser (1969)* proposed subgen. *Leprocybe* M.M. Moser with sect. *Bolares* Kühner & Romagn. ex M.M. Moser, sect. *Brunneotincti* Kühner & Romagn. ex M.M. Moser, sect. *Leprocybe* M.M. Moser, sect. *Limonii* M.M. Moser ex Nezdojm. and sect. *Orellani* M.M. Moser. *Peintner, Moncalvo & Vilgalys (2004)* found subgen. *Leprocybe* is polyphyletic

Corresponding authors
Di Liu, liudi@ybu.edu.cn
Tie-Zheng Wei,
weitiezheng@163.com

based on molecular phylogenetic analysis, and a similar result was obtained by *Harrower et al. (2011)*. *Soop et al. (2019)* proposed two new sections within this subgenus, sect. *Persplendidi* Soop & Dima and sect. *Veronicae* Soop. *Ammirati et al. (2021)* proposed three new sections, sect. *Fuscotomentosi* Niskanen, Liimat. & Ammirati, sect. *Melanoti* Niskanen, Liimat. & Ammirati and sect. *Squamiveneti* Niskanen, Liimat. & Ammirati as well as 11 new species from North America based on morphological, ecological and molecular data. The article also validated sect. *Veneti* Bellanger, Niskanen, Ammirati & Liimat., which was proposed by *Konrad & Maublanc (1937)*, and designated neo- or epitypes for four species. *Bidaud et al. (2021)* reported 11 species of the subgenus (included three new species) from the Mediterranean and confirmed 23 synonymies. *Liimatainen et al. (2022)* re-defined subgen. *Leprocybe* based on result from analysis of five genes. Seven sections, sect. *Fuscotomentosi*, sect. *Leprocybe*, sect. *Melanoti*, sect. *Persplendidi*, sect. *Squamiveneti*, sect. *Veneti* and sect. *Veronicae*, were included in the subgenus, sect. *Limonii* was placed in a new genus, *Aureonarius* Niskanen & Liimat.

Only three members of *Leprocybe* were reported in China. *Keissler & Hohwag (1937)* firstly reported *C. cotoneus* from Yunnan Province. *Shao & Xiang (1997)* documented *C. venetus* from Heilongjiang Province of Northeast China. *Xie (2022)* reported *C. nigrosquamosus* Hongo from Yunnan. In a survey of *Cortinarius* species in southwest China, two undescribed species of *Leprocybe* were found based on morphological and phylogenetic analyses. These species are described below with photos of morphological and microscopical characteristics.

## MATERIALS AND METHODS

### Morphological study

Specimens were collected from the Xizang Autonomous Region, Sichuan and Yunnan Provinces, China. The fresh basidiomata were photographed after collecting from the field and the macro-morphological characters were recorded in detail before drying in an oven at 45 °C. A 20% KOH solution was used on fresh pileus and stipe surface, lamella, and context for chemical reaction. Observation of basidiomata was performed under ultraviolet light at a wave length of 360 nm. The specimens were deposited in Fungarium, Institute of Microbiology, Chinese Academy of Sciences (also as Herbarium Mycologicum Academiae Sinicae, HMAS). Descriptions of the microscopical characters are from dried collections. Thin sections were prepared by hand with a razor blade. Sections were mounted in 5% KOH solution. Basidiospores, basidia, tramal hyphae, context, and pileipellis of pileus and stipe were measured using an ocular micrometer. At least 30 basidiospores and 20 basidia of each mature collection were measured.

### DNA extraction, amplification and sequencing

Genomic DNA was extracted from dried specimens using standard protocol (*Rogers & Bendich, 1994*). The DNA extracts were used as templates for PCR. Amplification reactions were performed to obtain sequences of nuclear ribosomal internal transcribed spacer region (ITS) using primer pairs ITS5/ITS4 (*White et al., 1990*). The amplification was carried out under the following conditions: initial denaturation for 3 min at 95 °C,

**Table 1 ITS sequences of subgen. *Leprocybe* used in phylogenetic analysis.**

| Name | Voucher | Locality | GenBank ID | References |
|---|---|---|---|---|
| *Cortinarius clandestinus* | SMI200A | Canada | FJ039583 | *Harrower et al. (2011)* |
| *C. clandestinus* | SAT03-137-02 | USA | FJ717552 | *Harrower et al. (2011)* |
| *C. clandestinus* | SMI24 | Canada | FJ157136 | *Harrower et al. (2011)* |
| *C. clandestinus* | JFA10285 | USA, Washington | MW009201 | *Ammirati et al. (2021)* |
| *C. cotoneus* | CFP1032 (neotype) | Sweden | MW009216 | *Ammirati et al. (2021)* |
| *C. cotoneus* | PML5260 | France | MW010117 | *Bidaud et al. (2021)* |
| *C. cotoneus* | PML5429 | France | MW010116 | *Bidaud et al. (2021)* |
| *C. cf. cotoneus* | HMAS260331 | China, Jilin | KX513578 | *Bidaud et al. (2021)* |
| *C. cf. cotoneus* | HMAS254210 | China, Sichuan | KX513580 | *Bidaud et al. (2021)* |
| *C. cf. cotoneus* | QL0601 | China | HM105543 | *Bidaud et al. (2021)* |
| *C. cf. cotoneus* | ZWL560 | China | KX444284 | *Bidaud et al. (2021)* |
| *C. flavifolius* | TENN068695 (epitype) | USA, Tennessee | MW009217 | *Ammirati et al. (2021)* |
| *C. flavifolius* | MICH256 | USA, Iowa | MW009238 | *Ammirati et al. (2021)* |
| *C. flavifolius* | T. Niskanen14-227 | USA, Tennessee | MW009218 | *Ammirati et al. (2021)* |
| *C. fuscoflavidus* | JFA11644 (holotype) | USA, Oregon | MW009221 | *Ammirati et al. (2021)* |
| *C. fuscoflavidus* | T. Niskanen09-158 | USA, Washington | MW009222 | *Ammirati et al. (2021)* |
| *C. fuscoflavidus* | DBB41055 | USA, California | MW009223 | *Ammirati et al. (2021)* |
| *C. hengduanensis* | HMAS250455 (holotype) | China, Yunnan | KX513581 | * |
| *C. hengduanensis* | HMAS145537 | China, Yunnan | KX513582 | * |
| *C. hengduanensis* | HMAS250509 | China, Yunnan | KX513583 | * |
| *C. hengduanensis* | HMAS272520 | China, Xizang | OR538887 | * |
| *C. hengduanensis* | HMAS270305 | China, Xizang | OR538888 | * |
| *C. hughesiae* | JFA13086 (holotype) | USA | MW009224 | *Ammirati et al. (2021)* |
| *C. hughesiae* | TENN068689 | USA, Tennessee | MW009225 | *Ammirati et al. (2021)* |
| *C. leproleptopus* | R. Henry8409 (holotype) | France | MW009226 | *Ammirati et al. (2021)* |
| *C. leproleptopus* | ST40 | Italy | MW010181 | *Bidaud et al. (2021)* |
| *C. leproleptopus* | AB 08-10-395 | France | MW010171 | *Bidaud et al. (2021)* |
| *C. leproleptopus* | GS1 | Germany | MW010092 | *Bidaud et al. (2021)* |
| *C. lutescens* | f1781 (holotype) | USA, New York | MW009228 | *Ammirati et al. (2021)* |
| *C. lutescens* | H7000893 | Canada, Newfoundland & Labrador | MW009229 | *Ammirati et al. (2021)* |
| *C. melanotus* | PML5454 | France | MW010120 | *Breitenbach & Kränzlin (2000)* |
| *C. melanotus* | CFP1101 | France | MW009230 | *Breitenbach & Kränzlin (2000)* |
| *C. melanotus* | PML5454 | France | MW010120 | *Breitenbach & Kränzlin (2000)* |
| *C. melanotus* | CFP1101 | France | MW009230 | *Breitenbach & Kränzlin (2000)* |
| *C. pescolanensis* | MCVE29054 (holotype) | Italy | NR_153070 | *Picillo & Marchionni (2016)* |
| *C. pescolanensis* | BP13/291 | Italy | KX010945 | *Picillo & Marchionni (2016)* |
| *C. pescolanensis* | JB8114/13 | Spain | KY657256 | *Ballara, Mahiques & Garrido-Benavent (2017)* |
| *C. pescolanensis* | PML5448 | France | MW010139 | *Bidaud et al. (2021)* |
| *C. selinolens* | MPU1116858 (holotype) | France | MW010172 | *Bidaud et al. (2021)* |
| *C. selinolens* | FR2013185 | Tunisia | MW010072 | *Bidaud et al. (2021)* |

(Continued)

| Table 1 (continued) | | | | |
|---|---|---|---|---|
| **Name** | **Voucher** | **Locality** | **GenBank ID** | **References** |
| *C. subcotoneus* | PML2143 (holotype) | France | MW010122 | *Bidaud et al. (2021)* |
| *C. subcotoneus* | AB08-10-331 | France | MW010167 | *Bidaud et al. (2021)* |
| *C. subcotoneus* | GS15 | Germany | MW010106 | *Bidaud et al. (2021)* |
| *C. veneto-occidentalis* | T. Niskanen11-051 (holotype) | USA, Alaska | MW009243 | *Ammirati et al. (2021)* |
| *C. veneto-occidentalis* | T. Niskanen11-281 | Canada, Alberta | MW009248 | *Ammirati et al. (2021)* |
| *C. veneto-occidentalis* | T. Niskanen11-258 | USA, Alaska | MW009245 | *Ammirati et al. (2021)* |
| *C. venetus* | CFP112 (neotype) | Sweden | MW009250 | *Ammirati et al. (2021)* |
| *C. venetus* | PC245 | France | MW009252 | *Ammirati et al. (2021)* |
| *C. venetus* | AB12-09-62 | France | MW010178 | *Bidaud et al. (2021)* |
| *C. venetus* | GH20100927 | Germany | MW010090 | *Bidaud et al. (2021)* |
| *C. venetus* | GS13 | Germany | MW010104 | *Bidaud et al. (2021)* |
| *C. cf. venetus* | HMAS274611 | China, Sichuan | KX513584 | *Bidaud et al. (2021)* |
| *C. cf. venetus* | HMAS274352 | China, Sichuan | KX513585 | *Bidaud et al. (2021)* |
| *C. cf. venetus* | HMAS268596 | China, Sichuan | KX513586 | *Bidaud et al. (2021)* |
| *C. cf. venetus* | 2M06 | Japan | LC373240 | *Bidaud et al. (2021)* |
| *C. yadingensis* | HMAS254811 | China, Sichuan | OR538889 | * |
| *C. yadingensis* | HMAS280697 | China, Sichuan | OR538890 | * |
| *C. yadingensis* | HMAS280698 | China, Sichuan | OR538891 | * |
| *C. yadingensis* | HMAS254819 (holotype) | China, Sichuan | OR538892 | * |
| *C. yadingensis* | HMAS254820 | China, Sichuan | OR538893 | * |
| *C. veronicae* | PDD68468 (holotype) | New Zealand | KC017355 | *Bidaud et al. (2021)* |
| *C. veronicoides* | MEL2120747 | Australia | GQ890324 | *Danks, Lebel & Vernes (2010)* |

**Note:**
* Means from this study.

followed by 30 cycles, at 95 °C for 30 s, at 55 °C for 45 s, at 72 °C for 60 s, and a final elongation step at 72 °C for 10 min.

## Phylogenetic analysis

The newly generated ITS sequences were submitted to GenBank. The ITS sequences for the phylogenetic analyses were selected based on results of BLASTn (>94% identity) in GenBank. Two species, *C. veronicae* (KC017355) and *C. veronicoides* (GQ890324), were chosen as outgroup. Seventy-three sequences (Table 1) were aligned and edited with BioEdit 7.2.2 (*Hall, 1999*). Bayesian inference (BI) and maximum likelihood (ML) methods were implemented to analyses in this study. MrModeltest 2.3 was used to calculate the best model (HKY+I+G) for BI analysis (*Nylander, 2004*). The BI analysis was performed with MrBayes 3.2.6 (*Ronquist & Huelsenbeck, 2003*), and the ML analysis was conducted in MEGA X (*Kumar et al., 2018*). The matrix contained 75 ITS sequences with 681 nucleotide sites is available from GenBank (see Data Availability section). Trees were viewed in FigTree 1.4.4 and processed in Adobe Photoshop 2019.
## Nomenclature

The electronic version of this article in Portable Document Format (PDF) will represent a published work according to the International Code of Nomenclature for algae, fungi, and plants, and hence the new names contained in the electronic version are effectively published under that Code from the electronic edition alone. In addition, new names contained in this work have been submitted to Index Fungorum from where they will be made available to the Global Names Index (https://gni.globalnames.org/). The unique Index Fungorum number can be resolved and the associated information viewed through any standard web browser by appending the Index Fungorum number contained in this publication to the prefix "https://www.indexfungorum.org/Names/Names.asp".
The online version of this work is archived and available from the following digital repositories: PeerJ, PubMed, Central SCIE, and CLOCKSS.

# RESULTS

## Molecular phylogeny

The ITS matrix for phylogenetic analyses included 73 sequences, representing 15 species. The resulting alignments were deposited at TreeBASE (http://www.treebase.org; submission ID 30908; accessed on 1 November 2023). The BI and ML trees showed similar topologies, and the ML tree was selected as the representative phylogeny (Fig. 1). The independences of *Cortinarius hengduanensis* and *C. yadingensis* were well- supported by phylogenetic analyses. *Cortinarius hengduanensis* has a close relationship with *C. flavifolius* Peck, *C. yadingensis* nests in a clade of sect. *Leprocybe* and clusters together with *C. contoneus*, *C. hughesiae* Ammirati, Matheny, Liimat. & Niskanen, *C. selinolens* Bidaud & Bellanger and *C. subcotoneus* Bidaud. The BLASTn against GenBank and UNITE databases taked the holotype specimensas the examples. The percent identity of *C. hengduanensis* with *C. flavifolius* (TENN068695, epitype) and *C. leproleptopus* (R. Henry8409, holotype) are 95% and 94%, respectively. The ITS sequence of *C. yadingensis* (HMAS254819, holotype) has 97%, 97%, 96% identity with *C. cotoneus* (CFP1032, neotype), *C. subcotoneus* (PML2143, holotype) and *C. hughesiae* (JFA13086, holotype), respectively.

## Taxonomy

***Cortinarius hengduanensis*** P. Hong, Ke Wang, Z. Du, M.L. Xie, Di Liu & T.Z. Wei, *sp. nov*. Fig. 2.

    **Index Fungorum:** IF901373.

    **Holotype:—CHINA, Yunnan Province**, Shangri-La County, Haba Mountains Nature Reserve, alt. 3,000 m, in mixed forest with *Pinus densata* and *Quercus aquifolioides*, 16 Aug. 2008, T.-Z. Wei, X.-Q. Zhang & F.-Q. Yu 194, HMAS 250455 (GenBank ID: KX513581 ITS).

    **Etymology:—**"hengduan", Chinese, referring Hengduan Mountains of southwest China, the locality of the type collection.

    **Diagnosis:—***Cortinarius hengduanensis* has olive tinged basidiomata, squamulose pileus, and small subglobose to broadly ellipsoid basidiospores (5.5–7.5 × 5–6.5 μm).

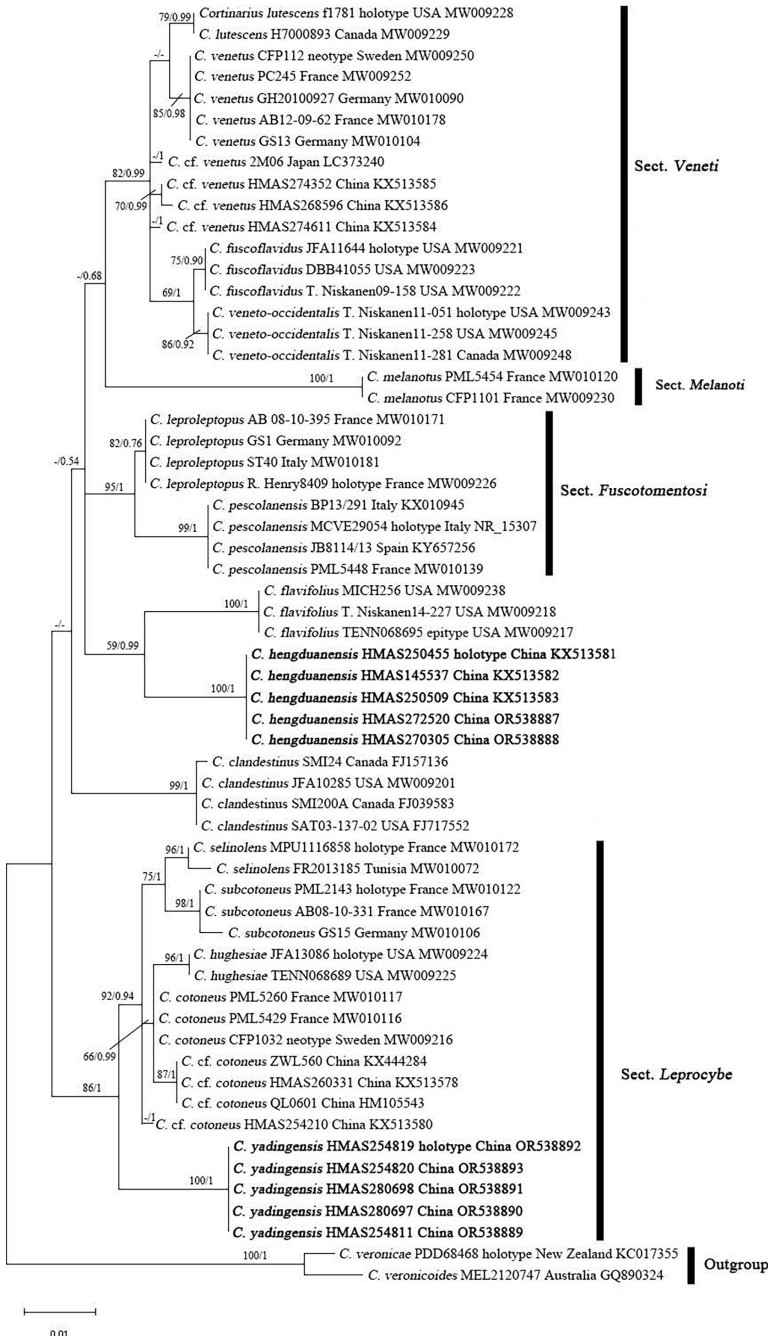

**Figure 1 ML phylogram inferred from the ITS dataset of *Cortinarius* species.** The ML bootstrap values (ML) ≥50% and Bayesian posterior probabilities (BPP) ≥0.95 are shown on the branches (BPP/ML). New species are marked in black bold font.

It differs from *C. flavifolius* for having greenish tint on pileus surface and differs from *C. venetus* by its obviously annulate stipe. ITS sequences of the new species (GenBank ID: KX513581–KX513583, OR538887 and OR538888) deviate from that of *C. flavifolius* by at least 28 substitutions and indel positions. In mixed forest with *Pinus yunnanensis*, *P. densata* and *Quercus aquifolioides*.

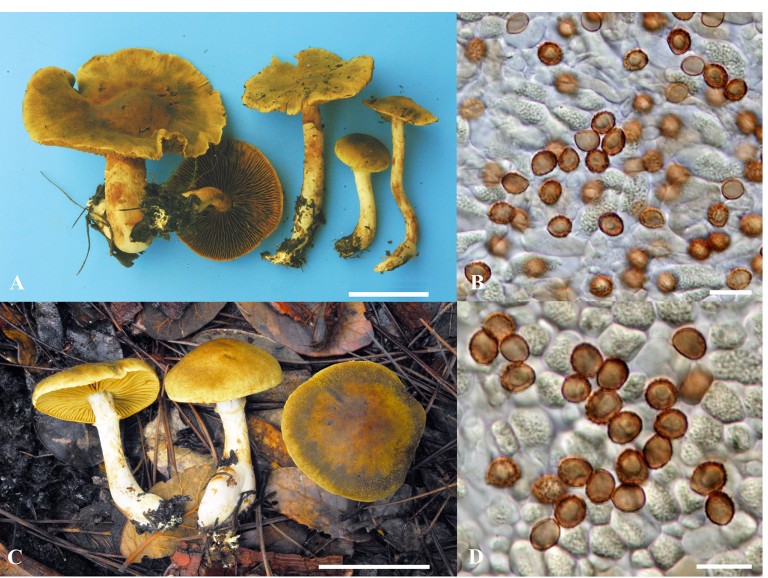

**Figure 2** *Cortinarius hengduanensis.* (A and B) HMAS 250455 (holotype). (A) Basidiomata (scale bar = 5 cm); B. basidiospores (scale bar = 10 μm). (C and D) HMAS 250509. (C) Basidiomata (scale bar = 5 cm); D. Basidiospores (scale bar = 10 μm).

**Description:**—*Pileus* 4.5–12.5 cm diam., hemispherical at first, later becoming convex to applanate, with a shallow, obtuse umbo when mature, margin sometimes uplifted or undulate, surface uneven, tomentose, persistently covered with small olive-brown (RAL 8008) to dark olive-brown (RAL 8022) squamules, margin radially striate, olive-green (RAL 6003) with brownish (RAL 8024) tint at first, then brownish olive (RAL 8000), olive-brown (RAL 8008) to dull brown (RAL 8019) with paler margin, sometimes darker at the center. *Context* up to 8 mm thick at pileus center, whitish (RAL 9010) to pale when dry, olive-gray (RAL 7015) when moist, fleshy, soft; odor indistinct, taste mild. *Lamellae* adnate to emarginate, 3–8 mm wide, close to moderately crowded, olive to grayish green (RAL 6006) when young, brown with olive (RAL 8000) tint to rust-brown (RAL 8012) when mature, edges paler. *Stipe* 5.5–12 × 0.7–2.6 cm, cylindrical or base enlarged and up to 3 cm in diam., surface completely covered with universal veil at first, then longitudinally fibrillose above and with conspicuous annular girdles below; olivaceous yellow (RAL 1020), tinged rust-brown (RAL 8012) from basidiospores, at first solid, then soft to hollow. *Partial veil* arachnoid, greenish to olive (RAL 6003), then becoming rust brown (RAL 8012) from mature basidiospores. *Universal veil* distinct and persistent, fibrillose, covering whole basidiomata at first, then forming fibrils and squamules on pileus and fibrils and girdles on stipe surface, pale with olive tint at first, brownish with olive (RAL 8000) tint to olive-brown (RAL 8008) when mature. *Basidiospore deposit* rust brown (RAL 8012). *Chemical reaction with 20% KOH* dark reddish brown on pileus and lamella, brown on stipe and reddish brown on context. *Fluorescence reaction under ultraviolet light* distinctly bright yellow on lamellae, and weaker on surfaces of pileus and stipe.

*Basidiospores* (5.2–) 5.5–7 (–7.5) × (4.8–) 5–6 (–6.5) μm, Q = (1.08–) 1.1–1.2 (–1.23) (av. = 1.15), subglobose to broadly ellipsoid, yellowish brown, moderately to distinctly

verrucose. *Basidia* 28–35 × 8–10 μm, clavate, thin-walled, mostly subhyaline, with four sterigmata. *Lamella edges* heterogeneous, with sterile cells, 20–30 × 6–9 μm, clavate, subhyaline, thin-walled. *Pleurocystidia* absent. *Subhymenial layer* up to 10 μm thick, of narrow and branched hyphae, hyaline, thin-walled, 2–4 μm diam. *Hymenophoral trama* regular, 80–100 μm wide, of hyaline and thin-walled hyphae, 5–20 μm diam. *Pileipellis*: epicutis well developed, hyphae 9–15 μm wide, subcylindrical, colourless, yellowish to brownish, thin-walled, smooth; *Hypodermium* present, hyphae 3–8 μm wide, irregular, with yellowish intracellular pigment in 5% KOH. Hyphae of the cortina 3–6 μm diam., subhyaline to yellowish, thin-walled. *Clamp connections* present.

**Additional Specimens Examined: CHINA, Yunnan Province**, Songming County, Aziying Town, alt. 2,000 m, in mixed forest with *Pinus yunnanensis* and *Quercus aquifolioides*, 5 Aug. 2005, T.-Z. Wei & F.-Q. Yu Gm1082, HMAS 145537 (GenBank ID: KX513582); Deqin County, Baima Mountains, alt. 3,100 m, in mixed forest with *P. densata* and *Q. aquifolioides*, 19 Aug. 2008, T.-Z. Wei, X.-Q. Zhang & F.-Q. Yu 258, HMAS 250509 (GenBank ID: KX513583); **Xizang**, Nyingchi City, Bayi District, Bayi Town, 29°38′02.58″ N, 94°23′44.72″E, alt. 3,360 m, in mixed forest with *P. densata* and *Q. aquifolioides*, 14 Sept. 2014, T.-Z. Wei, J.-Y. Zhuang, X.-Y. Liu & H. Huang 5281, HMAS 270305 (GenBank ID: OR538888); Lulang Town, alt. 3,481 m, 22 Sept. 2014, W.-L. Lu & Q.-M. Wang, 3017, HMAS 272520 (GenBank ID: OR538887).

**Notes:** *Cortinarius hengduanensis* is characterized by its distinct olive tinged basidiomata, fibrillose-squamulose pileus and subglobose to broadly ellipsoid basidiospores (5.2–7.5 × 4.8–6.5 μm). The new taxon is close to *C. flavifolius*, the latter also has a fibrillose to squamulose pileus and subglobose basidiospores (*Ammirati et al., 2021*). *Cortinarius flavifolius* is distributed in eastern North America. It does not have any olive or greenish tint on the pileus, and its basidiospores are 6.7–8.9 × 4.8–5.9 μm (*Ammirati et al., 2021*) larger than that of *C. hengduanensis*. The new species shows high morphological similarity to *C. venetus*, the latter has obvious olive basidiomata, subglobose basidiospores and distributed in Europe of mixed forest (*Breitenbach & Kränzlin, 2000*; *Bidaud et al., 2005*; *Soop, 2018*). The stipe of *C. venetus* is fibrillose when mature (*Breitenbach & Kränzlin, 2000*; *Bidaud et al., 2005*), and only with ephemeral veil annulate zone (*Bidaud et al., 2005*) or thin girdle (*Soop, 2018*) when young. However, the stipe surface of *C. hengduanensis* is obviously annulate and banded from persistent veil remains. *Cortinarius melanotus* Kalchbr. also has olive or olive tinged basidiomata and fibrilloso-squamulose pileus (*Breitenbach & Kränzlin, 2000*), but differs from the subglobose basidiospores of *C. hengduanensis*, its basidiospores are ellipsoid (6.2–8 × 4.3–5.3 μm) (*Breitenbach & Kränzlin, 2000*).

*Cortinarius yadingensis* P. Hong, Ke Wang, Z. Du, M.L. Xie, Di Liu & T.Z. Wei, *sp. nov.* Fig. 3.

**Index Fungorum:** IF901374.

**Holotype:—CHINA, Sichuan Province**, Daocheng County, Yading Scenic Spot, alt. 4,034 m, in mixed forest of *Abies* sp., *Larix potaninii*, *Picea* sp. and *Q. aquifolioides*, 18 Aug. 2016, T.-Z. Wei, L.-H. Sun, Z.-X. Wu & R.-C. Zhang, 7168, HMAS 254819 (GenBank ID: OR538892).

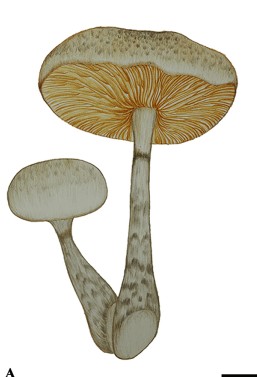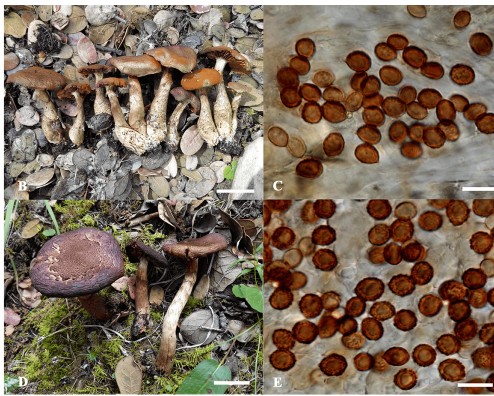

**Figure 3 *Cortinarius yadingensis*.** (A) Basidiomata (scale bar = 1 cm). (B and C) HMAS 254819 (holotype). (B) Basidiomata (scale bar = 3 cm); (C) Basidiospores (scale bar = 10 μm). (D and E) HMAS 254811. (D) Basidiomata (scale bar = 2 cm); (E) Basidiospores (scale bar = 10 μm).

**Etymology:**—"yading", Chinese, referring Yading Scenic Spot of Sichuan Province, China, the locality of the type collection.

**Diagnosis:**—*Cortinarius yadingensis* has distinct brown squamules on the basidiomata surface and subglobose to broadly ellipsoid basidiospores (6.5–9 × 6.3–7.6 μm). It differs from *C. cotoneus*, *C. hughesiae*, *C. selinolens* and *C. subcotoneus* by its lack of olivaceous coloration. ITS sequences of the new species (GenBank ID: OR538889–OR538893) are distinct from other members of sect. *Leprocybe* and deviating from them by at least 11 substitutions and indel positions. In mixed forest with *Abies* sp., *Larix potaninii*, *Picea* sp. and *Q. aquifolioides*.

**Description:**—*Pileus* 2–6 cm diam., hemispherical at first, later becoming convex to applanate, mostly with a shallow and obtuse umbo when mature, margin decurrent to straight, sometimes split when mature, surface pale brown (RAL 8025), grayish brown (RAL 8019) to rust brown (RAL 8012), covered brown (RAL 8000) to dark brown (RAL 8022) fibrillose squamules, radially striate, silky and shining. *Context* up to 5 mm thick at pileus center, pale brown (RAL 8025) to grayish brown (RAL 8019), fleshy, soft; odor indistinct, smell mild. *Lamellae* emarginate-adnate, up to 5 mm wide, brownish (RAL 8000) when young, rust-brown (RAL 8012) when mature, moderately crowded, edges margin paler. *Stipe* 3.5–8 × 0.5–1 cm, subcylindrical, with base up to 1.5 cm in diam., surface brownish (RAL 8000) to pale brown (RAL 8025), covered with brown (RAL 8000) to dark brown (RAL 8022) fibrillose squamules, with annular zone from partial veil, longitudinally striate; at first solid then soft to hollow. *Partial veil* arachnoid, grayish (RAL 7035) when young, forming a rust brown (RAL 8012) ring from basidiospores when mature. *Universal veil* forming fibrillose brown (RAL 8000) to dark brown (RAL 8022) squamules on the surface of the stipe. Basidiospore deposit rust brown. *Chemical reaction with 20% KOH* dark brown to blackish brown on pileus, lamella and stipe surface, and reddish brown on context. *Fluorescence reaction under ultraviolet* light bright yellow on lamellae, and weaker on surfaces of pileus and stipe.

*Basidiospores* (6.5–) 7–8.5 (–9) × (6.3–) 6.5–7.2 (–7.6) μm, Q = (1.03–) 1.08–1.2 (–1.24) (av. = 1.16); subglobose, yellow-brown, distinctly verrucose. *Basidia* 28–35 × 8–10 μm, clavate, thin-walled, mostly subhyaline, with four sterigmata. *Lamella edges* heterogeneous, with sterile hyphae, 20–30 × 6–9 μm, clavate, subhyaline, thin-walled. *Pleurocystidia* absent. *Subhymenial layer* up to 10 μm thick, of narrow and branched hyphae, hyaline, thin-walled, 2–4 μm diam. Hymenophoral trama regular, 80–100 μm wide, of hyaline and thin-walled hyphae, 5–20 μm diam. *Pileipellis*: epicutis well developed, hyphae 9–15 μm wide, subcylindrical, colourless, yellowish to brownish, thin-walled, smooth; *Hypodermium* present, hyphae 3–8 μm wide, irregular, with yellowish intracellular pigment in 5% KOH. *Hyphae of the cortina* 3–6 μm diam., subhyaline to yellowish, thin-walled. *Clamp connections* present.

**Additional Specimens Examined: CHINA, Sichuan**, Litang County, road side to Daocheng County, alt. 3,937 m, in conifer forest with *Abies* sp. and *Picea* sp., 17 Aug. 2016, T.-Z. Wei, L.-H. Sun, Z.-X. Wu & R.-C. Zhang, 7117, HMAS 254811 (GenBank ID: OR538889); Daocheng County, Yading Scenic Spot, alt. 4034 m, in mixed forest of *Abies* sp., *Larix potaninii*, *Picea* sp. and *Q. aquifolioides*, 18 Aug. 2016, T.-Z. Wei, L.-H. Sun, Z.-X. Wu & R.-C. Zhang, 7169, HMAS 254820 (GenBank ID: OR538893); 7123, HMAS 280697 (GenBank ID: OR538890); 7127, HMAS 280698 (GenBank ID: 254819).

**Notes:** *Cortinarius yadingensis* is characterized by its grayish brown squamulose to brown pileus and stipe and subglobose, distinctly verrucose basidiospores (6.5–9 × 6.3–7.6 μm). The species is phylogenetically close to four species of sect. *Leprocybe*, *C. cotoneus* (*Breitenbach & Kränzlin, 2000*, *Soop, 2018*), *C. hughesiae* Ammirati, Matheny, Liimat. & Niskanen (*Ammirati et al., 2021*), *C. selinolens* Bidaud & Bellanger (*Bidaud et al., 2021*) and *C. subcotoneus* (*Bidaud et al., 2005*). Compared with *C. yadingensis*, all the four related species have similar finely tomentose to squamulose pileus and strongly verrucose basidiospores. Unlike *C. yadingensis*, all the related species have olive tinged basidiomata. Besides, *C. cotoneus*, *C. selinolens* and *C. subcotoneus* inhabit in broad-leaf forest associated with Fagaceae trees, but *C. yadingensis* are reported in mixed forest with *Abies* sp., *Larix potaninii*, *Picea* sp. and *Q. aquifolioides*. Phylogenetically, *C. yadingensis* can be distinguished from its sister species by at least 11 substitutions and indels in their full ITS sequences.

## DISCUSSION

According to *Bidaud et al. (2021)*, *Ammirati et al. (2021)* and *Liimatainen et al. (2022)*, subgen. *Leprocybe* was re-defined as *Cortinarius* species with small- to medium-sized (occasionally large-sized) basidiomata, obvious UV fluorescent reaction, tomentose to squamulose pileus and subglobose basidiospores, characteristics found in both *C. hengduanensis* and *C. yadingensis*. *Cortinarius hengduanensis* has olive coloration, which was also found in most of the species following: sect. *Fuscotomentosi*, sect. *Leprocybe*, sect. *Melanoti*, sect. *Squamiveneti* and sect. *Veneti* (*Bidaud et al., 2021*; *Ammirati et al., 2021*). *Cortinarius yadingensis* does not display any olive tint, its pileus and stipe surfaces are covered by densely dark brown fibrillose squamules, which are found in a few *Leprocybe* species lacking olive coloration, such as *C. pescolanensis* (*Picillo &*

*Marchionni, 2016*). However, phylogenetic analyses of the present research does not support close relationship between *C. yadingensis* and *C. pescolanensis*. Ecologically, the two new species are reported in subalpine and alpine areas of southwest China mainly associated with *Picea* sp. and *Quercus* sp., but the related members of *Leprocybe* are mostly distributed in North America or Europe.

Compared with the known diversity of *Leprocybe* in North America and Europe (*Bidaud et al., 2021*; *Ammirati et al., 2021*), only three species were previously reportedfrom China. *Cortinarius cotoneus* was reported by *Keissler & Hohwag (1937)* in Yunnan according to a collection of 1914. *Horak (1987)* reexamined the specimen and confirmed its morphological identification. *Shao & Xiang (1997)* reported *C. cotoneus* in Heibei, Shaanxi, Sichuan, Guangdong and *C. venetus* in Heilongjiang,but these are lacking specimen citations. Sequences from materials of China and East Asia belonging to this lineage with doubtful annotations in GenBank were also included in the phylogenetic analysis (GenBank IDs: HM105543, KX444284, KX513578, KX513580, KX513584-KX513586, LC373240). These sequences nest near *C. cotoneus* and *C. venetus* in the phylogenies and may thus represent some unknown species in the section. Unfortunately, the sequenced Chinese materials (*e.g.*, HMAS260331, HMAS254210, HMAS274611, HMAS274352 and HMAS268596) are dried specimens without any *in situ* photos, which cannot provide sufficient morphological characteristics for the introduction of new species. *Xie (2022)* reported *Cortinarius nigrosquamosus* and identified it as a member of *Leprocybe* foryellowish to olivaceous basidiomata with black squamules. Although the ITS sequence of the studied specimens is the same, there are some morphological variations among the three specimens. Due to these unanswered questions on all those three *Leprocybe* species historically reported in China, more samplings are still needed to accumulate sufficient data to clarify the diversity of *Leprocybe* in China.

## CONCLUSION

Based on morphological and molecular evidence, *Cortinarius hengduanensis* and *C. yadingensis*, were proposed in *Cortinarius* subgenus *Leprocybe*. Both species were reported in mixed forest of Southwest China. *Cortinarius hengduanensis* has distinct olive tinged basidiomata, squamulose pileus, and small, subglobose to broadly ellipsoid basidiospores. *Cortinarius yadingensis* has a squamulose pileus and stipe, subglobose to broadly ellipsoid, coarsely verrucose basidiospores. Further research on the *Leprocybe* diversity in China is needed for the likely occurrence of additional new species in the country.

## ACKNOWLEDGEMENTS

The authors are grateful to three anonymous reviewers for their constructive comments and suggestions on the manuscript. We thank to Dr. Li-Hua Sun, Mr. Zu-Xun Wu (Baotou Teachers' College) and Mr. Run-Chao Zhang (China Agricultural University) for their kind help in fieldwork. We also thank to Ms. Jing Yang (Institute of Microbiology, Chinese Academy of Sciences) and Mr. Yao-Bin Guo (Shenyang Agricultural University) for their kind help in molecular phylogenetic studies.

### Funding

This work was supported by the National Key Basic Research Special Foundation of Chisna (2013FY110400), the National Key Research and Development Program of China (2017YFD0300104), the National Natural Science Foundation of China (No. 31270072) and the Special Funds for the Young Scholars of Taxonomy of the Chinese Academy of Sciences (No. ZSBR-001). The funders had no role in study design, data collection and analysis, decision to publish, or preparation of the manuscript.

### Grant Disclosures

The following grant information was disclosed by the authors:
National Key Basic Research Special Foundation of China: 2013FY110400.
National Key Research and Development Program of China: 2017YFD0300104.
National Natural Science Foundation of China: 31270072.
Young Scholars of Taxonomy of the Chinese Academy of Sciences: ZSBR-001.

### Competing Interests

The authors declare that they have no competing interests.

### Author Contributions

- Peng Hong performed the experiments, prepared figures and/or tables, authored or reviewed drafts of the article, and approved the final draft.
- Ke Wang performed the experiments, analyzed the data, prepared figures and/or tables, authored or reviewed drafts of the article, and approved the final draft.
- Zhuo Du performed the experiments, prepared figures and/or tables, contributed specimens, and approved the final draft.
- Ming-Jun Zhao performed the experiments, prepared figures and/or tables, drew the illustration, and approved the final draft.
- Meng-Le Xie conceived and designed the experiments, authored or reviewed drafts of the article, and approved the final draft.
- Di Liu conceived and designed the experiments, authored or reviewed drafts of the article, and approved the final draft.
- Tie-Zheng Wei conceived and designed the experiments, analyzed the data, authored or reviewed drafts of the article, and approved the final draft.

### Data Availability

The sequences are available at NCBI: KX513578, KX513580 to KX513586 and OR538887 to OR538893.

### New Species Registration

The following information was supplied regarding the registration of a newly described species:

Index Fungorum:

Cortinarius hengduanensis: https://www.indexfungorum.org/Names/NamesRecord. asp?RecordID=901373.

Cortinarius yadingensis: https://www.indexfungorum.org/Names/NamesRecord.asp? RecordID=901374.

## Supplemental Information

Supplemental information for this article can be found online at http://dx.doi.org/10.7717/ peerj.17599#supplemental-information.

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
