# Peer review of "Two new Cortinarius species in subgenus Leprocybe from Southwest China"

_PeerJ, doi:10.7717/peerj.17599_

## Round 0.1 · original submission · Major Revisions

Three experts assessed your manuscript and found that communicates a relevant study but some improvements are required. In particular, the improvement of Figure 2 and the discussion, along with better English usage.

**Language Note:** The Academic Editor has identified that the English language must be improved. PeerJ can provide language editing services - please contact us at [email protected] for pricing (be sure to provide your manuscript number and title). Alternatively, you should make your own arrangements to improve the language quality and provide details in your response letter. – PeerJ Staff

·

Basic reporting

This is a well structured paper. On the pdf I reviewed, I entered several corrections which are related to writing for the most part. I do think the introductory paragraph to the paper should review the current status of the Cortinariaceae in terms of genera, host plant, etc. The descriptions are very thorough and generally well written, including all necessary information for studies of Leprocybe. Illustrations are adequate and the phylogenetic tree is well structured. Overall thorough job. Methods, Materials and Literature review are good.

Experimental design

Not applicable.

Validity of the findings

Describing these two new species greatly extends our understanding of the subgenus Leprocybe.

Additional comments

None!

·

Basic reporting

The authors describe two new Cortinarius species of subgenus Leprocybe from China, based on morphological and molecular (ITS) data. Although the descriptions and phylogenetic analyses are within the international standards in the field of fungal taxonomy and the topic sufficiently introduced and referenced, this manuscript suffers from serious weaknesses, that should prevent it to be published in its current form.
The biggest issue is the quality of illustrations. The in situ photos of C. yadingensis (Fig. 3A and C) show old, dried out or maybe even frozen specimens that do not properly support the species description beyond the bulbous base and squamous surface of the stipe. The two photos of C. hengduanensis (Fig. 2A and C) are comparatively much better but the specimens in panel 2C are overexposed (flashed ?), masking stipe ornamentation. None of these four photos does show the context of basidiomata, which is often a crucial feature in Cortinarius. Overall, this part of the manuscript does not comply with the « exemplary production quality » that PeerJ includes in its highest standards for publication (Aims & Scope). Another in situ photograph of at least C. yadingensis should be provided to fix this issue.
Second, in its current format, the manuscript does not conform to the structure recommended by PeerJ, because it lacks a conclusion. Formally, such an issue should not be too difficult to fix by adding a few summarizing lines but then, the next weakness (the Discussion) will become even more apparent. The acknowledgments are also not conform with PeerJ recommendations, that exclude funding from this section.
Third, in taxonomic papers, the Discussion of the results often constitutes the real added-value of the work. Unfortunately, this part of Hong et al. falls short. In eleven lines, the authors state that their two spp. nov. fit in the current definition of subgenus Leprocybe, and they comment on their colour and veil remnants. Cortinarius (and not only this genus) diversity in China is a black box for most taxonomists of the world, so discussing the expected or suspected diversity of Leprocybe in China would have been most welcome here. Several Chinese sequences belonging in this lineage and with doubtfull annotations exist in Genbank, some included in past studies on the subgenus (e.g. HM105543, KX444284, KX513578, KX513580, KX513584-KX513586, LC373240 from Japan, Bidaud et al., 2021). Why were they ignored by the authors ? Instead, Hong et al. preferred to oversample in their phylogeny some very well resolved species from Europe or North America (e.g. C. leproleptopus, C. pescolanensis, C. clandestinus), for which a single sequence would have been sufficient. Including Chinese sequences unasigned to a known species in Fig. 1 would have dispensed the reader to check whether they may represent one of the two spp. nov. (they do not). Together with some comments about the ecology of C. yadingensis and C. hengduanensis, they may have also sparked interesting discussion about Leprocybe diversity in China.
Fourth, at the end of their Introduction, the authors mention findings of C. cotoneus, C. venetus and C. nigrososquamosus from China, but without any comment on the reliability of their taxonomic identity. Were these collections sequenced ? What is the evidence that the latter name belongs to subgenus Leprocybe ? This information is important at this place in the manuscript because it follows a paragraph on recent studies all including molecular results, but it would also feed the Discussion of Leprocybe diversity in China (see above).
Fifth, voucher ids in Table 1 and Fig. 1 could be usefully edited as to specify which sequences represent types. This has been done for some species but not for all (CFP1032, JFA 11644, JFA 13086, R. Henry 8409, f1781, MPU1116858, T. Niskanen 11-051 and CFP112).
Lastly, the English language should be improved. Examples of grammatical errors include line 52 (« is » missing before « distinguished »), lines 64, 66, 73, 80 (« based » should replace « base »), or line 162 that is hardly meaningful. Together with other minor issues (typos, spaces, errors in references…), these incorrect or unclear parts of the manuscript are highlighted in the attached annotated pdf file.

Experimental design

The submission describes original primary research within the Aims & Scope of the Journal. It clearly defines the research question, which is relevant and meaningful. The knowledge gap being investigated is real but may be more explicitely introduced and discussed. The investigation has been conducted rigorously and to acceptable technical standards, except for the illustrative parts. Methods are described with sufficient information to be reproducible by another investigator.

Validity of the findings

The data is robust, statistically sound, and controlled. It is provided in an acceptable discipline-specific repository. The conclusions are not specifically included as a section but as inferred from the Results, they are connected to the original question investigated, and limited to those supported by them.

Reviewer 3 ·

Basic reporting

The reviewer has attached detailed suggestions with the revised file uploaded. The authors are suggested to follow the suggestions.

Experimental design

no comment

Validity of the findings

no comment

Additional comments

no comment

Annotated reviews are not available for download in order to protect the identity of reviewers who chose to remain anonymous.

---

## Round 0.2 · Major Revisions

The manuscript was revised by Reviewers participating in the first round of assessment. Even though they appreciate the effort in improving the manuscript, they still detect relevant issues that need more elaboration, including discussion and the inclusion of images of better quality. The English usage was improved but still requires proofreading.

·

Basic reporting

I went through the word file and made additional track changes in it, they will show my name and are in blue.

Basically, the reported species, descriptions, and phylogeny are suitable. Discussion of the literature and related information needs additional work. I can look at this again if necessary.

Experimental design

OK!

Validity of the findings

Valid!

Additional comments

See manuscript track changes.

·

Basic reporting

The authors describe two new Cortinarius species of subgenus Leprocybe from China, based on morphological and molecular (ITS) data. Although the descriptions and phylogenetic analyses are within the international standards in the field of fungal taxonomy and the topic sufficiently introduced and referenced, this manuscript suffers from poor quality of some illustrations, that should prevent it to be published in its current form. The in situ photos of C. yadingensis (Fig. 3A and C) indeed show old, dried out or maybe even frozen specimens that do not properly support the species description beyond the bulbous base and squamous surface of the stipe. The two photos of C. hengduanensis (Fig. 2A and C) are comparatively much better but the specimens in panel 2C are overexposed (flashed ?), masking stipe ornamentation. None of these four photos does show the context of basidiomata, which is often a crucial feature in Cortinarius. Overall, this part of the manuscript does not comply with the « exemplary production quality » that PeerJ includes in its highest standards for publication (Aims & Scope). Another in situ photograph of at least C. yadingensis should be provided to fix this issue.
The English language should be improved, especially in the Discussion.

Experimental design

The submission describes original primary research within the Aims & Scope of the Journal. It clearly defines the research question, which is relevant and meaningful. The knowledge gap being investigated is real. The investigation has been conducted rigorously and to acceptable technical standards, except for the illustrative parts. Methods are described with sufficient information to be reproducible by another investigator.

Validity of the findings

The data is robust, statistically sound, and controlled. It is provided in an acceptable discipline-specific repository. The conclusions are connected to the original question investigated, and limited to those supported by them.

Reviewer 3 ·

Basic reporting

The reviewer have revised the manuscript. The English writing, date, and discussion have been improved.

Experimental design

No obvious problem about experimental design.

Validity of the findings

There is no doubt that these species are new to science.

Additional comments

The reviewer have attached the suggestions in detail with the files uploaded.

Annotated reviews are not available for download in order to protect the identity of reviewers who chose to remain anonymous.

---

## Round 0.3 · Minor Revisions

The reviewers assessed your manuscript. They appreciate the effort in improving the readability of the text. However, the manuscript has room for improvement in the English usage. The authors are advised to be assisted by a professional service in English editing.

**Language Note:** The Academic Editor has identified that the English language must be improved. PeerJ can provide language editing services - please contact us at [email protected] for pricing (be sure to provide your manuscript number and title). Alternatively, you should make your own arrangements to improve the language quality and provide details in your response letter. – PeerJ Staff

·

Basic reporting

See revised manuscript!

Experimental design

Not applicable.

Validity of the findings

Important.

Additional comments

See revised manuscript!

·

Basic reporting

See previous comments. The authors did not fully adressed my major concern targeting the poor quality of the in situ photograph but I understand that they have no other ones and I appreciate their inclusion of an additional drawing of C. yadingensis. The English has been improved, although some parts need further editing:
- Whole ms: check the spelling of your first sp. nov. "hengduanensis" or "henduanensis" ?
Also, "subgloboid" vs "subglobose", please use a single word consistently throughout the manuscript (I think subglobose is correct)
- Abstract: change the last sentence by "Both novelties have been found in mixed forests of southwest China"
- Results, Molecular phylogeny: correct "C. contoneus"
- C. hengduanensis Notes: correct "fi-brillose", "gir-dle", "sub-globose"
- Discussion: replace "Cortinarius yadingensis does not has any olive tint" by "Cortinarius yadingensis does not display any olive tint"
- Discussion: replace "(GenBank ID: HM105543, KX444284, KX513578, KX513580, KX513584-KX513586, LC373240), these sequences formed multiphyletic clades near C. cotoneus and C. venetus, which may represent some unknown new species." by "(GenBank ID: HM105543, KX444284, KX513578, KX513580, KX513584-KX513586, LC373240). These sequences nest near C. cotoneus and C. venetus in the phylogenies and may thus represent some unknown species in the section."
- Discussion: rephrase the following sentence, that is unclear: "Although the ITS sequence of the studied specimens is the same, there are some morphological variations among the three specimens"
- Conclusion: Trim and edit the last sentence as : "Further research on the Leprocybe diversity in China is needed for the likely occurrence of additional new species in the country."

Experimental design

No comment.

Validity of the findings

No comment.

Reviewer 3 ·

Basic reporting

The English writing of the manuscript has been improved in two round of reviewing. Tense consistency should be noticed. The authors are suggest to check the whole manuscript.

Detailed geographic information are needed in table 1 for those countries that have vast territories, if available.

Numbers of the subgen. Leprocybe members in China or Asia can be provided.

Experimental design

Standard color code should be used in morphological description.

It is better to use pileipellis and stipitipellis, because “cutis” is often described as a form of repent cuticular layer.

The standard applied in the best model selection (AIC or hLRT) should be notified.

Validity of the findings

ITS blasting results can be provided in result part.

The two new species are reported in subalpine and alpine areas. This character can be discussed with the closely related taxa in note/discussion part.

Additional comments

The reviewer has commented the suggestions for improving the manuscript in detail with the PDF file uploaded. The authors are suggested to improve the manuscript following the comments and suggestions. Intensive checking of the whole manuscript for those same or similar improvements are needed.

Annotated reviews are not available for download in order to protect the identity of reviewers who chose to remain anonymous.

---

## Round 0.4 · accepted · Accept

The manuscript is now suitable for the next stage of editorial production.